# Timing Matters: A Systematic Review of Early Versus Delayed Palliative Care in Advanced Cancer

**DOI:** 10.3390/cancers17152598

**Published:** 2025-08-07

**Authors:** Ioana Creangă-Murariu, Eliza-Maria Froicu, Dragos Viorel Scripcariu, Gema Bacoanu, Mihaela Poroch, Mihaela Moscalu, Claudia Cristina Tarniceriu, Teodora Alexa-Stratulat, Vladimir Poroch

**Affiliations:** 1Advanced Research and Development Center for Experimental Medicine (CEMEX), 700454 Iasi, Romania; ioana.creanga@d.umfiasi.ro (I.C.-M.); teodora.alexa-stratulat@umfiasi.ro (T.A.-S.); 2Medical Oncology-Radiotherapy Department, “Grigore T. Popa” University of Medicine and Pharmacy, 700115 Iasi, Romania; 3Centre for Translational Medicine, Semmelweis University, 1085 Budapest, Hungary; 42nd Internal Medicine Department, “Grigore T. Popa” University of Medicine and Pharmacy, 700115 Iasi, Romaniavladimir.poroch@umfiasi.ro (V.P.); 5Department of Surgery, “Grigore T. Popa” University of Medicine and Pharmacy, 700115 Iasi, Romania; 6Department of Palliative Care, Regional Institute of Oncology, 700483 Iasi, Romania; 7Department of Preventive Medicine and Interdisciplinarity, “Grigore T. Popa” University of Medicine and Pharmacy, 700115 Iasi, Romania; 8Department of Morpho-Functional Sciences, “Grigore T. Popa” University of Medicine and Pharmacy, 700115 Iasi, Romania

**Keywords:** early palliative care, cancer care, quality of life, symptom burden

## Abstract

Many believe palliative care is only for end of life, but starting early can improve symptoms, quality of life, and even survival in advanced cancer. However, when and how to deliver it remain unclear. This review examines clinical trials to understand the true impact of early palliative care and what makes it most effective. The findings can guide better care, future research, and policies that support earlier integration into cancer treatment.

## 1. Introduction

Cancer has become a public health problem, especially in developing countries. An important increase in the worldwide number of cancer cases to 35.3 million by 2050 is anticipated, compared to the 2022 estimate of 20 million [1]. Cancer survivors undergoing active treatment face many physical and psychosocial problems, often endured together with their families, which contributes to a negative impact on quality of life and nevertheless overall survival [2].

Defined by the World Health Organization, palliative care is a method that enhances the quality of life for patients and their families who are confronted with the challenges associated with life-threatening illness [3]. The impact of cancer-related symptoms on individuals with advanced diseases is well recognized, as inadequate symptom control negatively affects multiple aspects of the patient’s overall illness experience. Comprehensive pain and symptom management is one of the fundamental principles of palliative care, which is designed to enhance the physical, emotional, and functional well-being of patients with advanced and incurable diseases [4].

Although traditionally recommended in advanced stages, where patients are close to end of life, recent research shows that adding palliative care (PC) early in the oncology care continuum is linked to better patient-reported outcomes. However, these two scenarios for applying palliative care might differ in clinical practice. Early palliative care (EPC) focuses on providing relief from symptoms and stress early in the course of a serious illness, with the goal of improving quality of life and potentially extending survival [5]. End of life care (EoL PC), on the other hand, is geared towards helping patients and their families cope with the physical, emotional, and spiritual challenges that come with the end stages of a disease [6].

Despite mounting evidence, clinical trials exhibit substantial heterogeneity in how and when early palliative care is implemented. Definitions of EPC vary widely; some interventions begin within 8 weeks of diagnosis, while others are based on symptom triggers or clinician judgment. Moreover, differences in healthcare system infrastructure, provider availability, and patient access contribute to inconsistent timing and delivery of EPC across settings [7].

The ultimate purpose of this review is to give clinicians clarifications regarding EPC by guiding them towards a more standardized, and ultimately more efficacious EPC integration. Our study’s aim is to examine the impact of EPC on key clinical outcomes in cancer patients, with a particular focus on its effectiveness in alleviating symptom burden, enhancing quality of life, and potentially influencing disease progression, by combining evidence from high-quality randomized clinical trials.

## 2. Materials and Methods

This systematic review was conducted to evaluate the impact of early integration of palliative care on clinically meaningful outcomes, specifically quality of life, symptom burden, and disease progression, focusing on adult patients with advanced malignancies. The review adhered to the Preferred Reporting Items for Systematic Reviews and Meta-Analyses (PRISMA) 2020 guidelines (Appendix A), and its methodology was guided by the recommendations outlined in the Cochrane Handbook for Systematic Reviews of Interventions [8]. The study protocol was developed in advance and prospectively registered on the PROSPERO international prospective register of systematic reviews (Registration ID: CRD42024623219), ensuring transparency and methodological consistency. Amendments were made: outcomes were reorganized as primary and secondary to ensure clarity and the planned GRADE assessment was not performed due to high study heterogeneity and the narrative nature of the synthesis.

### 2.1. Eligibility Criteria

We included only randomized controlled trials (RCTs) where the effects of early palliative care in adults diagnosed with advanced, incurable, or metastatic cancers were assessed. Eligible studies were required to evaluate a structured palliative care intervention introduced early in the disease trajectory, typically within eight weeks of diagnosis of advanced disease or at the initiation of systemic therapy. Interventions had to be multidisciplinary in nature and integrated with ongoing oncologic management.

To be considered for inclusion, studies needed to report on at least one of the following outcomes: overall quality of life (assessed using validated tools such as the EORTC QLQ-C30 or FACT-G), symptom burden (including physical and psychological symptoms such as pain, fatigue, or depression), or indicators of disease progression (such as time to progression or overall survival when relevant). The comparator in all studies had to be standard oncologic care, either with delayed palliative involvement or no formal palliative support. No time or language restrictions were applied.

Studies were excluded if they did not focus on populations with advanced-stage cancer, did not evaluate early palliative care interventions, or failed to include relevant outcome measures. Additionally, we excluded observational studies, non-randomized trials, case reports or case series, review articles, study protocols, conference abstracts, and studies for which the full text was not available.

### 2.2. Search Strategy

A comprehensive search of the literature, with no time restrictions, was carried out on 19th December 2024, across three major electronic databases: PubMed, Embase, and the Cochrane Central Register of Controlled Trials (CENTRAL). The search strategy was developed using both controlled vocabulary and free-text terms to capture the full scope of the relevant literature. The following combination of search terms was used: (Cancer OR Cancer* OR Neoplasia OR Neoplasm OR Neoplas* OR Tumor OR Tumors OR Tumour OR Tumours OR Malignancy OR Malignancies OR Malignan* OR Carcinoma OR Carcin* OR Oncolog* OR Adenocarcinoma OR Adenocarcin*)AND(incurable OR metastatic OR advanced)AND(early OR integrat* OR systematic)AND(palliative care OR palliation OR palliative medicine OR terminal care OR hospice care)AND(random*).

This search yielded a total of 4579 unique records, distributed as follows: 1411 articles from MEDLINE, 2264 from Embase, and 904 from CENTRAL. The reference lists of all included articles were further checked using *citationchaser* (Version 2.0, Stockholm Environment Institute, Sweden) [9] on 30 December 2024 to identify eligible articles, with no additional articles included.

### 2.3. Study Selection

All retrieved citations were imported into EndNote v21 reference management software (Clarivate Analytics) to identify and remove duplicates, resulting in a refined dataset of 2726 unique articles. Subsequently, two independent reviewers (IC-M, E-MF) performed a two-phase screening process. In the first phase, titles and abstracts were assessed for relevance to the inclusion criteria. Articles deemed potentially eligible were then subjected to full-text review. Cohen’s kappa coefficient (κ) was 0.82; any disagreements regarding inclusion were discussed and resolved through consensus or, when necessary, adjudicated by a third reviewer (GB).

During the full-text review phase, each article was scrutinized against the eligibility criteria outlined in the registered protocol. A total of 42 studies met all inclusion criteria and were retained for final analysis (Cohen’s kappa coefficient (κ), 1). Reasons for exclusion of studies during the screening process included the following: ineligible population (e.g., patients with early-stage cancer), absence of an early palliative care intervention, irrelevant or absent outcomes, inappropriate comparator groups, non-randomized design, duplicate publication, or unavailability of the full text.

### 2.4. Data Extraction and Management

Data extraction was conducted independently by two reviewers (IC-M, E-MF) using a predefined and pilot-tested data collection template to ensure consistency and minimize extraction bias. Each eligible study was reviewed in full, and relevant data were systematically recorded. Key bibliographic and methodological variables included the name of the first author, year of publication, and country in which the study was conducted. Detailed demographic and clinical information was extracted, including cancer type and staging, the operational definition of early palliative care as described in each study (e.g., timing from diagnosis, frequency of encounters, and multidisciplinary components), as well as how delayed or on-demand palliative care was defined in the comparator group. Symptom profiles were carefully extracted, including the specific symptoms assessed (e.g., pain, fatigue, dyspnea, anxiety) and the tools used for their evaluations, such as the Edmonton Symptom Assessment System (ESAS), Hospital Anxiety and Depression Scale (HADS), or other validated instruments. The duration of follow-up for each study population was also recorded.

The primary outcomes of interest were improvements in quality of life and symptom burden, as reported using standardized patient-reported outcome measures, along with any available data on disease progression or survival. Secondary outcomes included healthcare resource utilization metrics (e.g., emergency department visits, hospitalizations) and end-of-life outcomes such as place of death. Additional observations or noteworthy methodological features were also recorded.

Any inconsistencies in data abstraction between reviewers were resolved through discussion and consensus, with arbitration by a fourth reviewer (GB) when necessary. All extracted data were organized in structured tables and synthesized narratively.

### 2.5. Risk of Bias Assessment

Risk of bias assessment was independently conducted by two reviewers (IC-M and E-MF), with any discrepancies resolved by a third investigator (GB), performed with the Cochrane Risk of Bias 2 (ROB2) tool for RCTs [10].

## 3. Results

A total of 4579 records were identified through the systematic literature search. Following screening and eligibility assessment, 41 studies met the inclusion criteria and were included in the final analysis [11,12,13,14,15,16,17,18,19,20,21,22,23,24,25,26,27,28,29,30,31,32,33,34,35,36,37,38,39,40,41,42,43,44,45,46,47,48,49,50]. A summary of the selection process is presented in Figure 1.

### 3.1. Characteristics of Included Studies

Across the 41 randomized controlled trials included in this review, all studies recruited adult patients with advanced, metastatic, or incurable cancers, including but not limited to lung [12,13,14,26,27,40,41,47,48,49], gastrointestinal [11,13,14,26,32,38,40,41,47,49], breast [13,14,20,28,47,49], pancreatic [26,32,46], head and neck malignancies [36], or mixed [15,16,17,18,19,21,22,23,24,25,30,31,33,34,35,37,39,43,45,50]. Study settings were primarily hospital-based and conducted in high-income countries across North America [13,14,20,23,24,27,28,34,37,40,41,42,45,47,49,50], South America [17], Europe [11,15,19,25,26,29,30,32,33,38,39,43,44], and Asia [18,22,31,46,48], with the majority embedded in academic or tertiary cancer centers.

The definition of early palliative care (EPC) varied considerably in both timing and structure. The most common definition defined EPC by specific time thresholds, typically meaning the initiation of palliative care within 2 to 12 weeks of cancer diagnosis or treatment onset [11,14,22,26,27,30,31,32,36,38,41,43,44,48,51]. A second group of studies defined EPC through structured, recurrent follow-up, often involving monthly or biweekly visits with the palliative care team regardless of time since diagnosis [15,19,21,28,34,35,40,42,43,44,46,49]. A third model emphasized manualized or interdisciplinary interventions, such as ENABLE nurse-led coaching, STEP Care protocols, or stratified referral based on symptom burden [13,14,17,20,34,37,39,47]. Lastly, other studies employed pragmatic or flexible definitions, initiating EPC in response to clinical triggers or loosely structured around surgical or inpatient events [19,23,25,29,35,37,45].

Comparators were uniformly standard oncology care with delayed, non-protocolized, or on-demand palliative involvement. Most studies used validated, patient-reported outcome measures to evaluate primary endpoints such as quality of life and symptom burden, including instruments like the EORTC QLQ-C30, FACT-G, Edmonton Symptom Assessment System (ESAS), and Hospital Anxiety and Depression Scale (HADS). Several trials also assessed secondary endpoints such as survival duration, place of death, and healthcare utilization, including emergency visits and hospitalizations. Table 1 summarizes the key study characteristics including intervention type, outcomes, and follow-up duration.

Early palliative care was provided with consistent and clinically meaningful benefits in patients with advanced cancer in 32 of the included studies. Early palliative care was associated with significant improvements in patient-reported quality of life, with multiple studies reporting higher scores on validated instruments such as the EORTC QLQ-C30, FACT-G, and McGill Quality of Life Questionnaire. Moreover, symptom burden, including pain, fatigue, anxiety, depression, and dyspnea, was significantly reduced in the early palliative care arms. Interventions that included regular, structured symptom assessments and psychosocial support were particularly effective in controlling physical and emotional symptoms, suggesting that the timing and consistency of supportive care delivery play a critical role in alleviating distress in this population.

Although EPC was associated with improved outcomes in most studies, several trials did not demonstrate statistically significant benefits compared to standard or delayed care. Some studies reported no meaningful changes in global quality of life or psychological distress between intervention and control groups [25,49]. In other cases, primary outcomes such as symptom burden, emotional well-being, or overall patient satisfaction remained comparable between arms despite regular palliative follow-up [15,26,35,39]. A few trials found that EPC benefits were confined to specific subpopulations, such as those with greater baseline symptom burden or particular tumor types, while no significant effects were seen in the broader study cohort [37,42]. Some interventions that relied on proactive referral strategies or symptom-triggered models similarly failed to produce improvements in primary endpoints such as quality of life or functional well-being [29,33]. In at least one large randomized trial, the EPC intervention did not lead to significant changes in global health status as measured by validated quality of life scales [11]. One feasibility study highlighted logistical barriers to implementing structured EPC across multiple cancer types and was not powered to assess clinical effectiveness [20].

While overall survival was not the primary endpoint in most trials, several studies reported modest but statistically significant survival benefits among patients receiving early palliative care [12,14,41]. These findings challenge the misconception that palliative care is solely end-of-life care and supports its integration alongside curative or disease-modifying treatments. Furthermore, improved survival was not achieved at the expense of increased healthcare utilization or aggressive care at the end of life; in fact, several trials reported reduced rates of emergency department visits, hospital admissions, and ICU stays, suggesting that early palliative care enhances care coordination and anticipatory planning [25,41,44,49].

Place of death and care alignment with patient preferences were also more favorable in the early palliative care groups. Patients in these arms were more likely to die at home or in hospice, in line with stated preferences, and less likely to receive high-intensity interventions in the final weeks of life [14,39,41]. This shift reflects the ability of early palliative interventions to facilitate timely goals-of-care discussions and advanced care planning.

Importantly, we found significant variability in how early and delayed palliative care were defined and implemented across studies. While most interventions were multidisciplinary and integrated into oncology clinics, the timing, frequency, and content of palliative visits differed. Nonetheless, studies with more intensive or frequent palliative involvement generally reported more pronounced effects, supporting a dose–response relationship [31,35,41,46].

### 3.2. Risk of Bias

Risk of bias was assessed in all included articles which revealed several recurring methodological limitations that warrant careful considerations (Figure 2). Although most trials demonstrate low risk in domains such as randomization and outcome measurement, concerns frequently arose in relation to intervention blinding, outcome subjectivity, or missing data. Unblinded intervention delivery was a common issue, often paired with the use of subjective, patient-reported outcomes, increasing the risk of performance and detection bias. These vulnerabilities are particularly significant in palliative care research, where outcomes like symptom burden, emotional well-being, and quality of life are inherently subjective and sensitive to both participant and researcher expectations [52].

Additional concerns included high attrition rates in some trials, which introduce uncertainty regarding the completeness and representativeness of outcome data. In certain studies, the proportion of participants lost to follow-up was high, raising the possibility of bias due to differential dropout. Some trials also lacked sufficient clarity regarding outcome reporting, suggesting potential selective reporting or insufficient pre-specification of analyses. While a subset of studies exhibited consistently low risk across all domains, the cumulative presence of even isolated methodological flaws in others calls for cautious interpretation of overall findings. 

## 4. Discussion

This systematic review of 41 RCTs reinforces and expands upon the substantial body of evidence demonstrating that EPC is a clinically valuable intervention for individuals with advanced cancer. Across a diverse spectrum of malignancies, healthcare systems, and palliative care delivery models, EPC consistently resulted in improvements in QoL, reductions in symptom burden, and, though reported less frequently, was associated with prolonged survival and more appropriate end-of-life care.

Improved outcomes were most consistently observed in articles where attention was offered to all aspects of EPC, including multidisciplinary interventions. The key to improvement is based on early initiation of palliative care, typically within the first 2–4 weeks of diagnosis, with regular, structured follow-ups, often on a monthly basis. Moreover, programs that incorporated a multidisciplinary team and provided psychosocial and emotional support, such as counseling, coping strategies, and family engagement, demonstrated the greatest benefits in quality of life, mood, and symptom burden.

In the majority of included trials, EPC demonstrated a clear and measurable improvement in QoL, as assessed by validated instruments such as the EORTC QLQ-C30, FACT-G, and the McGill Quality of Life Questionnaire. Landmark studies, including Temel et al. (2010) [41], revealed both survival benefits and significant enhancements in QoL and mood among patients with metastatic non-small-cell lung cancer receiving EPC. Similarly, Vanbutsele et al. (2018) [43] and Zimmermann et al. (2014) [49] highlighted notable emotional and functional improvements, particularly when EPC was proactively structured and fully integrated into standard oncology workflows. Another consistent benefit was the alleviation of physical and psychological symptom burden, with reductions in pain, dyspnea, fatigue, and depression being frequently observed. This was evident in studies by Scarpi et al. (2019) [38], Kang et al. (2024) [31], and Chen et al. (2022) [18], in which proactive symptom monitoring and timely interdisciplinary interventions were identified as key contributors to symptom control.

Several trials in our review reported improvements in survival and healthcare utilization with EPC, though the sustainability and economic implications of these benefits remain incompletely understood. Temel et al. (2010) [41] demonstrated a median survival benefit of 2.7 months in metastatic NSCLC alongside reduced aggressive end-of-life care, and ENABLE III [14] observed a trend toward longer survival (14.0 vs. 8.5 months) with early versus delayed EPC, though this result was not statistically significant. Trials assessing healthcare utilization [38,49,50] consistently showed numerical reductions in hospitalizations, emergency department visits, and ICU admissions, but effects were often modest and did not reach statistical significance, reflecting heterogeneous intervention intensity and follow-up duration.

Cost-effectiveness and long-term sustainability remain critical gaps. Economic evaluations are limited but suggest potential cost savings through reduced acute care use: Vanbutsele et al. (2018) [43] reported fewer hospital deaths and lower end-of-life resource utilization, and Dionne-Odom et al. (2020) [34] highlighted reduced ICU use and hospitalizations, which could translate to meaningful cost offsets. However, few RCTs incorporate formal cost-effectiveness analyses or follow-up beyond 6–12 months, leaving the durability of survival and utilization benefits uncertain.

Beyond clinical symptom management, EPC was found to facilitate improved alignment between medical care and patients’ values and goals. Trials by Temel et al. (2010) [41], Greer et al. (2016) [27], and Zimmermann et al. (2014) [49] demonstrated that patients receiving EPC were more likely to avoid aggressive interventions near the end of life, less likely to be admitted to intensive care units, and more often died in preferred settings such as home or hospice. These outcomes reflect the positive impact of EPC on fostering timely and meaningful goals-of-care discussions and promoting dignified end-of-life experiences.

One key point of this review is that it synthesizes findings from a wide range of EPC interventions, including recent innovations such as the STEP Care model, ENABLE nurse-led interventions, and telehealth-enabled EPC programs. Through this analysis, a potential dose–response relationship emerged: interventions that were delivered more frequently, involved multidisciplinary teams, and adhered to structured protocols tended to yield superior clinical outcomes. For instance, trials incorporating sustained engagement, such as biweekly or monthly visits, demonstrated greater improvements in QoL, symptom control, and healthcare resource utilization compared to those employing one-off or loosely structured approaches, as illustrated in the studies by Eychmüller et al. (2021) [25] and Groenvold et al. (2017) [29].

Nonetheless, while 32 of the 41 trials reported significant clinical benefits, this review also underscores the notable heterogeneity and implementation challenges associated with EPC. Definitions of EPC varied widely: some studies introduced palliative care at predefined time points following diagnosis [32], others initiated care based on symptom burden [37], and some employed pragmatic or combined criteria, and was performed either by oncologist, the palliative care physician, or advanced practice nurses. Similarly, the comparator groups labeled as “standard” or “delayed” care encompassed a wide range, from complete absence of palliative input to reactive referrals, thereby complicating efforts to measure the true magnitude of EPC’s impact.

A major contribution of this review lies in the detailed mapping of EPC delivery characteristics, such as initiation timing, visit frequency, and team composition, against observed outcomes. Few previous analyses have systematically explored these implementation variables. Our synthesis reveals that not all EPC is equal: studies that employed structured, protocol-driven models consistently achieved better results, particularly in psychosocial and emotional domains (e.g., Kang et al. 2024 [18,31]. Conversely, trials in which EPC was delivered via ad hoc consultations or introduced too late in the illness trajectory tended to report weaker or no measurable benefits, as seen in the trials by Brims et al. (2019) [15] and Adenis et al. (2024) [11].

Among the included studies, thirty-two demonstrated significant improvements in at least one patient-centered outcome, whereas nine did not show measurable benefits in quality of life, symptom burden, or survival [11,15,25,26,29,33,35,36,38]. Neutral studies frequently shared several characteristics: they enrolled heterogeneous patients with very aggressive natural history (e.g., pleural mesothelioma or upper gastrointestinal malignancies), delivered minimal or single-contact interventions, relied on short follow-up durations (≤12 weeks), or employed narrow or insensitive primary endpoints. Additionally, control arms often provided enhanced supportive care, reducing the detectable treatment contrast. By comparison, the trials demonstrating benefits implemented early, structured, multidisciplinary EPC with monthly or protocolized follow-up and integrated psychosocial and family support, assessed with validated, sensitive longitudinal quality-of-life tools. The absence of benefit in the neutral trials likely reflects a combination of attenuated intervention intensity, insufficient observation windows to capture gradual improvement, and robust supportive care in the comparator arms.

Despite strong support from guidelines [53,54], the real-world adoption of EPC remains suboptimal. Many oncology practices continue to reserve palliative care for late-stage disease or imminent death, in part due to persistent misconceptions equating palliative care with end-of-life care. Additional barriers include inadequate provider training in palliative principles, insufficient institutional support, workforce shortages, and fragmented care infrastructures. Studies by Dionne-Odom et al. (2020) [34] and Dey et al. (2023) [22] reinforce the understanding that palliative care is not limited to symptom relief but also encompasses enhanced communication, shared decision-making, and emotional support for both patients and caregivers. Emerging delivery models, leveraging nurse navigators, telehealth technologies, or symptom-based triage, offer promising strategies to expand EPC access, particularly in underserved or resource-limited settings.

To translate these insights into clinical practice, EPC should be initiated at the time of advanced cancer diagnosis or the start of systemic therapy and delivered through structured, regular follow-ups, typically every two to four weeks, by a multidisciplinary team including physicians, nurses, psychologists, and social workers. Seamless integration into oncology workflows can be achieved through institutional care pathways, EMR-triggered referral systems, and structured patient education, while telehealth platforms, nurse navigator programs, and electronic symptom monitoring tools can expand access and enhance efficiency. Despite these strategies, widespread adoption remains limited by workforce shortages, insufficient clinician training, and persistent misperceptions that equate palliative care solely with end-of-life care, particularly in resource-constrained settings where infrastructure is underdeveloped. To overcome these barriers, streamlined referral systems, scalable and simplified delivery models, and robust institutional support are needed to achieve sustainable and equitable EPC implementation. Future research should prioritize the development of EPC models tailored to low-resource environments, conduct prospective cost-effectiveness analyses, and extend follow-up in RCTs and pragmatic trials to determine whether survival and healthcare utilization benefits are durable over time. Additionally, implementation science studies of scalable approaches, including nurse-led, telehealth, and EMR-based EPC, should evaluate long-term effects on patient outcomes, healthcare costs, and caregiver well-being, while integrating patient-reported and economic endpoints to capture the full health system value of EPC.

## 5. Conclusions

In summary, this review confirms that early palliative care consistently improves quality of life, reduces symptom burden, and supports patient-centered decision-making in advanced cancer. Its benefits are strongest when delivered early, regularly, and through multidisciplinary teams. To optimize impact, EPC should be integrated into routine oncology care as a structured intervention, not delayed or applied ad hoc. Future efforts must focus on overcoming implementation barriers and adapting effective models to diverse healthcare settings, ensuring broader and more equitable access to high-quality palliative care.

## Figures and Tables

**Figure 1 cancers-17-02598-f001:**
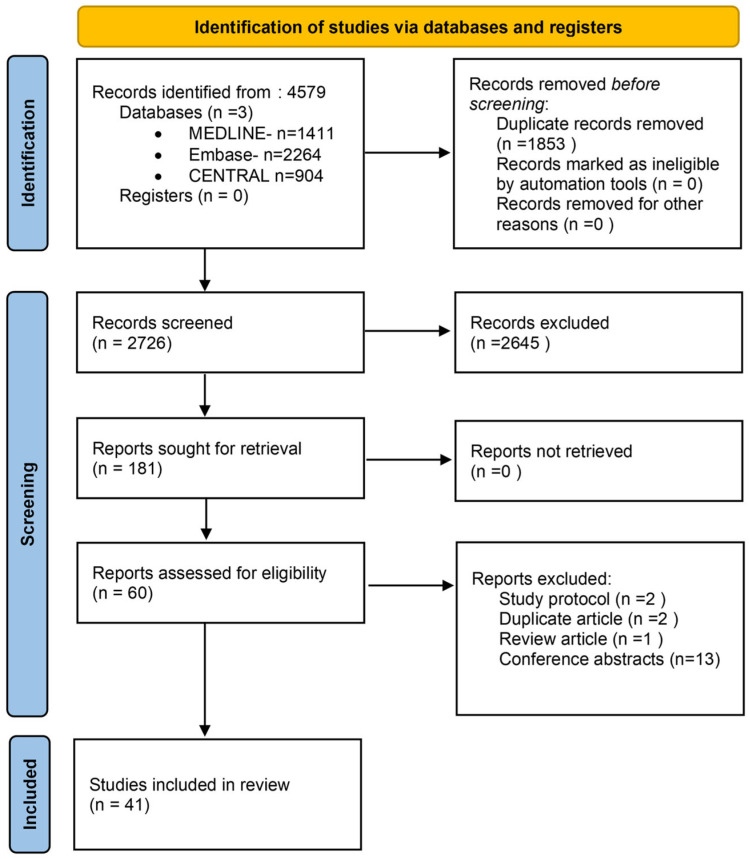
PRISMA flowchart of the article selection.

**Figure 2 cancers-17-02598-f002:**
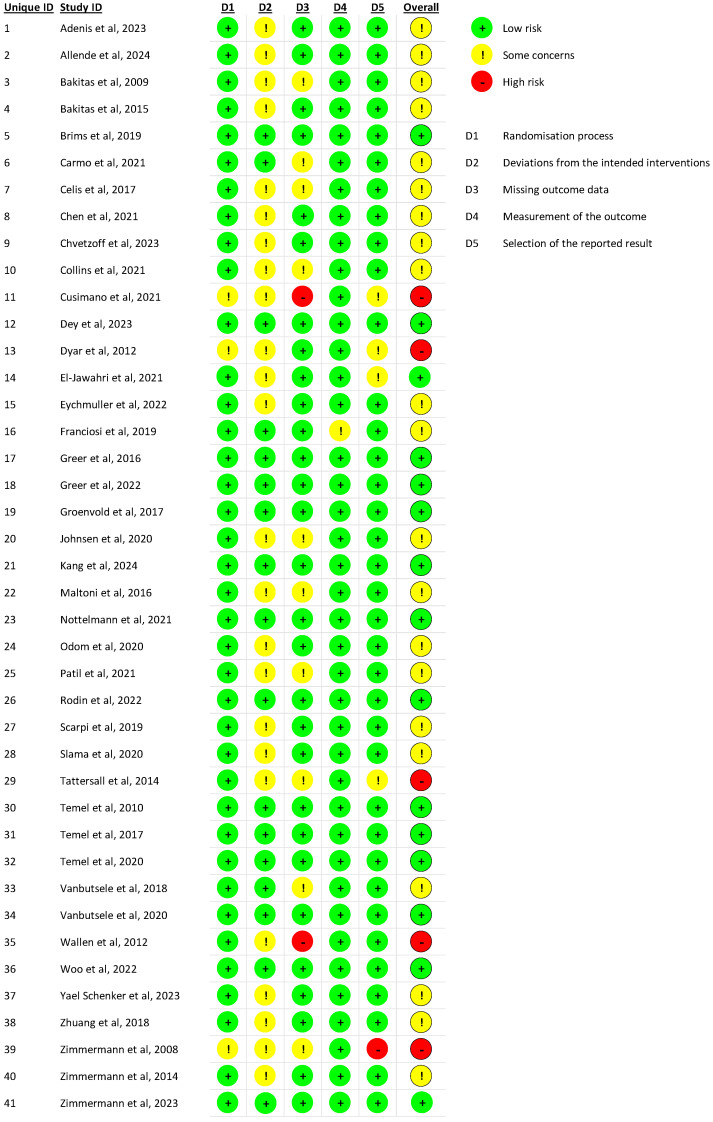
Risk of bias of included studies [11,12,13,14,15,16,17,18,19,20,21,22,23,24,25,26,27,28,29,30,31,32,33,34,35,36,37,38,39,40,41,42,43,44,45,46,47,48,49,50].

**Table 1 cancers-17-02598-t001:** Key characteristics of included studies.

Author, Year	Cancer Type	Stage	Definition of Early PC	Definition of Delayed PC	Assessed Symptoms	Method for Symptom Assessment	Follow-Up	Effective Intervention?	Results of Findings
Adenis et al., 2024 [11]	Upper GI cancers	Metastatic	PC integrated with oncology for 24 weeks	SOC alone	QoL, anxiety, depression	EORTC QLQ-C30, HADS	24 weeks	No	No difference in QoL or survival
Allende et al., 2024 [12]	NSCLC	Metastatic	Programmed visits to meet with the palliative, nutrition, and psychological care specialists at baseline and after the 2nd, 4th, and 6th cycles	PC assessment on request	QoL, physical, functional, emotional, social, lung symptom, anxiety, depression, survival	EORTC-QLQ-C30, HADS, ESAS	baseline, at 2nd, 4th, and 6th cycles of treatment	Yes	Better survival for baseline QoL values > 70, no significant changes in terms of QoL or symptom burden
Bakitas et al., 2009 [13]	GI, lung, genitourinary, breast	Unresectable or metastatic	4 weekly educational sessions (practical, emotional, spiritual, physical, family) + monthly follow-up	Standard oncology care, with access to PC services as needed	QoL, symptom intensity, mood, hospital/ICU visits	FACIT-Pal, ESAS, CES-D	1-month post-baseline, then every 3 months	Yes	Higher QOL and mood; no significant difference in symptoms or health care utilization
Bakitas et al., 2015 [14]	Lung, GI, breast, genitourinary, hematologic, other	Advanced	Initial PC consultation + 6 weekly APN calls + monthly follow-up	Same intervention initiated 3 months later	QoL, symptom impact, mood, survival, hospital use	FACIT-Pal, QUAL-E, CES-D	1 year	Yes	Improved 1-year survival, numerically fewer ED/hospital visits, not statistically significant
Brims et al., 2019 [15]	Pleural mesothelioma	Advanced	PC physician visit within 3 weeks; follow-up every 4 weeks for 24 weeks	Standard care; PC on request	QoL, symptom burden, mood, satisfaction	EORTC QLQ-C30	12 and 24 weeks	No	No difference between groups; preference for intervention noted
Carmo et al., 2017 [16]	Various solid tumors	Metastatic	Visits every 3 weeks post-randomization	Usual oncology care only	Depression, anxiety	PHQ-9, HADS	12 weeks	Yes	Reduced depression and anxiety
Celis et al., 2021 [17]	Various solid tumors	Advanced/metastatic	Patient navigation from baseline including multidisciplinary support	Usual oncological care with referral-based access to PC	Pain, fatigue, sleep disturbance, depression/anxiety	FACT-G, PHQ-2, G8, symptom-specific validated tools	12 weeks	Yes	Increased implementation of supportive care, more AD completion, less pain
Chen et al., 2021 [18]	Various solid tumors	Stage IV	Monthly PC and psychosocial support	Usual care	Depression, anxiety, QoL	HADS, FACT-G, ESAS	12 weeks	Yes	Improved depression, anxiety, QoL
Chvetzoff et al., 2022 [19]	Various solid tumors	Advanced/metastatic	Monthly PC clinic + symptom monitoring	Usual care	QoL, mood, satisfaction	EORTC, HADS, satisfaction scale	12 weeks	Yes	Improved QoL and patient engagement
Collins et al., 2022 [20]	Advanced breast, prostate, and brain	Advanced/metastatic	STEP Care: triggered referral at defined disease-specific transition points, min. monthly PC consultations for 3 months	Usual best practice cancer care; PC on referral	QoL, symptom distress, mood, illness understanding, service use	Patient/caregiver-reported outcomes, feasibility metrics, survival follow-up	Up to 36 months (brain); 24 months (breast, prostate)	Feasibility study	Trigger-based early PC was feasible in brain cancer but not breast/prostate; standardized triggers aid equity/access
Cusimano et al., 2023 [21]	Various solid tumors	Stage II–IV	PC within 4 weeks of enrollment	Usual care	QoL, symptom burden	FACT-O, PHQ-9	8 weeks	Yes	Feasible, acceptable, improved support
Dey et al., 2023 [22]	Cervical cancer	Locally advanced (IB2 to IIIB)	PC integrated from the start of chemoradiotherapy	Standard chemoradiotherapy alone	QoL subscales: physical, emotional, social, functional	FACT-G	3 months after treatment completion	Yes	Improved social and emotional well-being in intervention arm
Dyar et al., 2012 [23]	Various solid tumors	Advanced	Structured inpatient PC intervention	Usual care	Pain, fatigue, dyspnea	BPI, ESAS	6 weeks	Yes	Greater symptom improvement
El-Jawahri et al., 2016 [24]	Hematologic malignancies	Advanced	Initiated at hospital admission, frequent PC	Standard transplant care	QoL, depression, anxiety	HADS, FACT-BMT	100 days	Yes	Improved QOL and depression
Eychmüller et al., 2021 [25]	Various solid tumors	Advanced	Single structured PC session after randomization	Standard oncology care only	Distress, health-related QoL	NCCN distress thermometer, FACT-G	6 months	No	No significant differences in distress or QoL
Franciosi et al., 2019 [26]	NSCLC, gastric, pancreatic, biliary tract	Advanced	Systematic PC visits by physician/nurse team from baseline	PC on request	QoL	FACT-G	12 weeks	No	No significant improvement in QoL
Greer et al., 2022 [28]	Breast cancer	Advanced/metastatic	Monthly PC sessions from baseline	Standard oncology care	Depression, anxiety, coping	HADS, PHQ-9, PEACE Scale	24 weeks	Yes	Improved coping and emotional QOL
Greer et al., 2016 [27]	NSCLC	Advanced	Integrated outpatient PC from diagnosis	Standard oncology care only	QoL, healthcare costs	Cost analysis, clinical outcomes	Until death	Yes	Decreased costs
Groenvold et al., 2017 [29]	Various solid tumors	Advanced	Specialist PC team referral based on needs screening	Standard care	Primary need symptom score, QoL	EORTC QLQ-C30	3 and 8 weeks	No	No significant effect on primary or secondary outcomes
Johnsen et al., 2019 [30]	Various solid tumors	Advanced	PC consult + structured multidisciplinary follow-up	Usual oncology care only	QoL, mood, nausea	EORTC QLQ-C30, HADS	8 weeks	No	Slight nausea improvement only
Kang et al., 2024 [31]	Various solid tumors	Stage IV	Monthly PC + telecoaching from baseline	Usual oncology care	QoL, function	EORTC QLQ-C15-PAL, MQOL, SAT-SF	24 weeks	Yes	Improved emotional and physical functioning
Maltoni et al., 2016 [32]	Gastric and pancreatic cancers	Advanced	PC initiated at diagnosis, with monthly follow-up	PC on request	QoL, pain, nausea, anxiety, depression	EORTC QLQ-C15-PAL, HADS	3 months	Yes	Improved QoL and mood
Nottelmann et al., 2021 [33]	Various solid tumors	Advanced	PC within 2 weeks of diagnosis	Standard oncology care	QoL, symptom needs	EORTC QLQ-C30	12 weeks	No	No significant difference
Odom et al., 2019 [34]	Various solid tumors	Advanced	ENABLE: structured telehealth coaching by nurse for caregivers	Usual support	Caregiver QoL, depression, burden	CES-D, Zarit Burden Interview, CQOLC	24 weeks	Yes	Improved caregiver QoL and lower depression
Patil et al., 2021 [36]	Head and neck cancers	Locally advanced/metastatic	PC initiated before first chemotherapy cycle	Usual care	QoL, symptom burden	EORTC QLQ-C30	12 weeks	Yes	Significant improvement in QoL, reduction is hospital visits, not significant
Rodin et al., 2022 [37]	Various solid tumors	Advanced	PC from baseline stratified by symptom burden	PC upon request	QoL, symptom distress	ESAS, FAMCARE, FACT-G	12 weeks	Yes	Improved satisfaction and QOL in high-symptom group
Scarpi et al., 2023 [38]	Gastric cancer	Advanced	PC within 2 weeks of enrollment; 2–4 week follow-up	On request	QoL, anxiety, depression	FACT-Ga, HADS	12 weeks	Yes	Improved psychological outcomes, fewer ED/hospital visits, not significant
Slama et al., 2021 [35]	Various solid tumors	Advanced	PC every 6–8 weeks from inclusion	Standard oncology care	QoL, anxiety, depression	EORTC QLQ-C30, HADS	6 months	No	No significant difference in primary outcomes
Tattersall et al., 2014 [39]	Various solid tumors	Advanced	Structured question prompt list and early referral to PC team	Usual care	QoL, communication, unmet needs	POS consultation ratings	1 month	Yes	Improved communication and satisfaction
Temel et al., 2010 [41]	NSCLC	Metastatic	Initial PC within 3 weeks, then monthly	Standard oncology care	QoL, depression, survival	FACT-L, PHQ-9	12 weeks	Yes	Improved QoL, less depression, longer survival
Temel et al., 2017 [42]	GI and lung cancers	Advanced	PC within 4 weeks, monthly follow-up	Standard oncology care	QoL, mood	HADS, FACT-G, PHQ-9	24 weeks	Yes	Improved QOL and mood in lung subgroup, no significant reduction in ED visits or hospitalizations
Temel et al., 2020 [40]	GI and lung cancers	Advanced	Monthly structured visits with PC clinician	Usual oncology care only	QoL, coping, communication	FAMCARE, HADS, FACT-G	24 weeks	Yes	Improved coping, communication
Vanbutsele et al., 2018 [43]	Various solid tumors	Advanced	Structured monthly PC from enrollment	Usual care only	QoL, emotional well-being, satisfaction	EORTC QLQ-C30, CANHELP Lite	12 weeks	Yes	Improved emotional functioning and satisfaction
Vanbutsele et al., 2020 [44]	Various solid tumors	Advanced	Same model as 2018: early and systematic integration of PC with monthly visits	PC on request	QoL near end of life, healthcare use	EORTC QLQ-C30, McGill QOL	Until death	Yes	Better QOL scores 6, 3, and 1 month before death
Wallen et al., 2012 [45]	Surgical oncology patients	Advanced	Hospital-based PPCS starting post-surgery with interdisciplinary support	Standard surgical/oncologic care	Pain, symptom distress, mood, support, satisfaction	Gracely scales, Symptom Distress Scale, interviews	Up to 12 months	Yes (longer term)	Improved satisfaction, communication, symptom perception
Woo et al., 2019 [46]	Pancreatobiliary cancers	Advanced/metastatic	PC initiated within 8 weeks of diagnosis	Phone advice only	Pain, depression	BPI, CES-D	12 months	Yes	Improved pain control
Yael Schenker et al., 2023 [47]	GI, breast, lung	Advanced	Telehealth visits biweekly then monthly	PC on request	QoL, communication, emotional support	Caregiver CANHELP, PHQ-9, BPI	16 weeks	Yes	Improved caregiver satisfaction
Zhuang et al., 2018 [48]	NSCLC	Metastatic	Monthly visits from palliative team using standardized PC framework	Conventional oncology treatment only	QoL, depression, anxiety, pulmonary function	QOL Scale, HADS, PHQ-9, Pulmonary function indices	12 weeks	Yes	Improved QOL, mood, and lung function in early PC group
Zimmermann et al., 2014 [49]	Lung, GI, breast, others	Advanced/metastatic	PC visit within 4 weeks of enrollment; monthly thereafter	Usual care; PC if referred	Depression, anxiety, QoL	FACIT-Sp, QUAL-E, ESAS	4 months	Yes	Improved QoL, symptoms, patient satisfaction, trend toward fewer ED visits, not significant
Zimmermann et al., 2023 [50]	Various solid tumors	Advanced	Scheduled monthly PC visit from baseline	Usual care; no structured PC	Distress, QoL	ESAS-r, FACT-G	6 months	Yes	Lower symptom distress and better communication, no significant effect on ED visits or hospitalizations

Abbreviations: PC = palliative care, SOC = standard of care, QoL = quality of life, GI = gastrointestinal, NSCLC = non-small cell lung cancer, APN = advanced practice nurse, ICU = intensive care unit, ED = emergency department, EORTC-QLQ-C30 = European Organization For Research And Treatment of Cancer-Core Quality of Life questionnaire, HADS = Hospital Anxiety and Depression Scale, ESAS = Edmonton Symptom Assessment System, FACIT-Pal = Functional Assessment of Chronic Illness Therapy—Palliative Care; CES-D = Center for Epidemiologic Studies Depression Scale, QUAL-E = Quality of Life at the End of Life questionnaire, PHQ-9 = Patient Health Questionnaire-9, G8 = Geriatric Screening Tool, POS = Palliative Care Outcome Scale, FACT = Functional Assessment of Cancer Therapy, BPI = Brief Pain Inventory.

## Data Availability

This is a systematic review; no new data were created.

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
