# Peer review of "Timing Matters: A Systematic Review of Early Versus Delayed Palliative Care in Advanced Cancer"

_cancers, 2025, doi:10.3390/cancers17152598_

Round 1
Reviewer 1 Report
Comments and Suggestions for Authors
This is an interesting manuscript. I think it deserves to be published as it is.
Author Response
We thank the reviewer for the appreciation!
Reviewer 2 Report
Comments and Suggestions for Authors
This is a review of the article entitled “Timing Matters: A Systematic Review of Early Versus Delayed Palliative Care in Advanced Cancer.” Thank you for this opportunity to read this interesting article. There are several important points that authors should pay attention to.
- Authors used PRISMA 2020 flow diagram to show different phases of the systematic review. In PRISMA 2020, information from each database should be recorded separately in the diagram.
- Why did the authors limit themselves to only three databases and not use Web of Science and Scopus?
- Studies for which full text is available and which have been reviewed but excluded from the study should be listed in a Table with the author's name, article title, and reason for exclusion. This Table is displayed in the supplementary file.
- The time period for searching of studies should be written in the method section.
- Search strategy: In the search strategy section, duplicate words should be removed (Tumor or Tumors).
- This study has a protocol (https://www.crd.york.ac.uk/PROSPERO/view/CRD42024623219), and this is one of the advantages of a systematic review. But in some places of the protocol, information does not match with the article. Changes should be written in the “protocol amendments” in the method section of the article, for example: “Protocol-Cohen's kappa coefficient (κ) will be reported to assess inter-rater agreement during the selection process. Additional articles will be manually searched and identified (Small study) publication bias will be assessed by visual inspection of the funnel plots and (modified) Egger's test will be carried out if possible. The quality assessment of the included studies will be performed with GRADE-Pro, based on the recommendation of the Cochrane Collaboration.from the reference lists of primarily eligible studies.”
Primary and secondary outcomes were not written in the protocol.
- Table 1. Key characteristics of included studies: The full name of all abbreviations in the Table should be written in the Table footnote.
- Table 1: It is suggested that these words be revised in Table 1: “better with intervention?” and “observations.” They can revise to: “effective intervention” and “Results or Findings.”
Author Response
We thank the reviewer for the comments, we attached the response in a word document.

Reviewer 3 Report
Comments and Suggestions for Authors
The authors investigated the benefit of early palliative care for patients with advanced cancer by reviewing previous trials based on the timing of care. These findings can be helpful in improving the QOL and outcomes of patients in clinical practice. The reviewer raised the following concerns for improving the manuscript:
Regarding the heterogeneity of early palliative care (EPC) interventions, they varied considerably in terms of timing, frequency, and structure. How was this heterogeneity considered in the analysis and interpretation?
Based on the findings obtained, which specific components (e.g., frequency of visits, multidisciplinary approach, psychosocial support) seemed to be most consistently associated with improved outcomes?
Nine of the included RCTs did not demonstrate a significant clinical benefit from EPC. What were the main factors contributing to these results (e.g., patient selection, intervention intensity, and health system constraints)? These findings should be incorporated into the conclusions.
Given the limitations of subjective outcome measures, given that many primary outcomes (e.g., quality of life, symptom burden) relied on patient-reported measures, and that blinding was often not feasible, how did it mitigate potential performance or detection bias in the analysis? It should be considered that more objective endpoints (e.g., emergency department visits, hospitalizations, functional status) are integrated in the results.
While the evidence for early integration is strong, many institutions still struggle to adopt EPC due to limited resources, workforce shortages, and misperceptions that associate palliative care solely with end of life. It is better to add descriptions of specific strategies or policy recommendations that could improve EPC uptake and integration into oncology workflow.
Most included trials were conducted in high-income countries. How do authors view the applicability of these findings in LMIC settings, where access to palliative care may be limited? It could be valuable to develop simplified or tiered EPC models adapted to resource-constrained environments.
Several trials have suggested improvements in survival and healthcare utilization with EPC. Have there been any trends or gaps in the literature regarding the sustainability of these benefits and their cost-effectiveness? Please add the description if necessary. In addition, please discuss priorities for future research to evaluate the long-term value of EPC interventions.
Author Response

(The authors gave the same response as above.)
